# Adherence to Gluten-Free Diet in Coeliac Paediatric Patients Assessed through a Questionnaire Positively Influences Growth and Quality of Life

**DOI:** 10.3390/nu12123802

**Published:** 2020-12-11

**Authors:** Deianira Pedoto, Riccardo Troncone, Margherita Massitti, Luigi Greco, Renata Auricchio

**Affiliations:** 1Department of Medical Translational Sciences, University Federico II, 80131 Naples, Italy; troncone@unina.it (R.T.); marghym@gmail.com (M.M.); ydongre@unina.it (L.G.); r.auricchio@unina.it (R.A.); 2European Laboratory for the Investigation of Food-Induced Diseases, University Federico II, 80131 Naples, Italy

**Keywords:** coeliac disease, compliance, gluten free diet, diet adherence

## Abstract

Assessment of adherence to gluten-free diet (GFD) represents the cornerstone in the management of coeliac disease. The primary aim of this study was to assess diet adherence through a questionnaire adapted to children. The secondary aim was to identify influencing factors and outcomes related to diet adherence. In this study, data about diagnosis, education, quality of life (QoL) and anti-transglutaminase (anti-TG2) titers of 160 coeliac children were collected. For the assessment of diet adherence, all participants completed the questionnaire modified from Leffler et al. (2009), while a random sample of 37 also underwent an extensive dietary interview. According to the questionnaire, diet adherence was excellent in 95 (59.4%), fair in 46 (28.8%) and low in 19 (11.9%) patients. Children diagnosed with biopsy showed better adherence than those with a biopsy-sparing approach (*p =* 0.036). Adherence to GFD tended to worsen during the follow up, with the average length of follow up being associated with lower scores of diet adherence (*p =* 0.009). Moreover, adherence to GFD decreased throughout school career, dropping from elementary until high school (*p =* 0.037). A positive correlation was observed between adherence to GFD and growth percentiles, which increased when higher scores of adherence were achieved. Diet adherence positively correlated with QoL (*p =* 0.001). In conclusion, the questionnaire displayed good sensitivity in detecting problems in diet adherence, being useful as a screening tool. Better comprehension of influencing factors and outcomes may allow the development of new strategies to improve diet adherence.

## 1. Introduction

Nowadays, gluten free diet (GFD) represents the only therapeutic strategy for patients with coeliac disease (CD). Good adherence to GFD leads to the regression of clinical symptoms and intestinal mucosal damage, influencing prognosis in terms of morbidity and quality of life (QoL) [1]. Nevertheless, it requires a lifelong change in lifestyle, presenting a challenging task for CD patients. This problem is particularly relevant among children: it is estimated that up to one third of children fail to follow GFD [2], mainly adolescents, with significant consequences on growth and QoL [1].

Though the assessment of adherence to GFD represents the cornerstone in the management of coeliac patients, guidelines on this issue are notably lacking [3,4]. In clinical practice, adherence to diet is mainly evaluated by measuring anti-transglutaminase (anti-TG2) levels [5]. However, this method is not sensitive enough to detect infrequent gluten exposures [3,6,7]. On the other side, repeated duodenal biopsies to monitor mucosal recovery are not justified in children [8], since they involve unnecessary risks, representing a burden for patients and their families.

Despite the absence of a gold standard to assess diet adherence, a dietary evaluation by a trained dietitian is nowadays considered the method of choice [3,4,9]. Nevertheless, this method is not standardized and is time consuming in the clinical setting. For this reason, short and score-based standardized questionnaires have been recently developed for the assessment of compliance in adult patients [10,11], while few questionnaires are currently available for children [12], and nearly all of them derive from those validated for adults.

The difficulty in the assessment of adherence to GFD is increased by the interaction with several influencing factors, which need to be considered during the clinical evaluation [5], especially in children. Influencing factors may be divided into extrinsic (sociocultural background including parents’ education and job), related to the patient (age, gender, education, comorbidity), and related to the disease (age of diagnosis, diagnostic approach, length of follow up). The main outcomes which should be considered for children with CD are growth, QoL and nutritional parameters.

The primary aim of this study was to assess adherence to GFD using a questionnaire specific for Italian paediatric patients and compare this method with serology and dietary interview. The secondary aim was to identify factors which may be associated with compliance and outcomes (growth and QoL).

## 2. Materials and Methods

### 2.1. Patients and Visit

One hundred and sixty coeliac patients from southern Italy, 101 females (63.1%), 2–20 years old, mean age 11.7 ± 4.5, were consecutively admitted to this observational prospective study from May 2018 until June 2019. All subjects gave their informed consent for inclusion before they participated in the study. During the visit, data about socioeconomic background (parents’ grade of education and job) and patient (age, gender, education) were collected. Data about diagnosis such as age, diagnostic approach (biopsy or biopsy-sparing approach) and length of follow up were registered. Each patient underwent a complete anthropometric and physical examination. Parents gave their approval for participation in the study.

### 2.2. Assessment of Diet Adherence

Adherence to GFD was assessed through a questionnaire adapted from Leffler short questionnaire [10] and specifically modified for paediatric patients, which was administered by 2 independent clinicians to all the patients (Table 1). It consisted of 8 questions about perceived adherence to GFD, outdoor food consumption, GFD specific knowledge, self-efficacy and disease individual perception. Each answer was given a score (0–3). According to the score achieved, patients were classified in 3 groups of compliance: score ≤ 2 = excellent compliance, score 3–6 = fair compliance and score ≥ 7 = low compliance. Patients aging ≥ 12 years (*n* = 93) were asked to autonomously complete the questionnaire, guardian collaboration was allowed if required. For patients aging < 12 years (*n* = 67), guardians were asked to complete the questionnaire, while collaboration of the patient was always required. The results of the questionnaire were compared with anti-TG2 titers, performed by ELISA (Eurospital, Trieste, Italy) or by chemiluminescence (Delta) at time of follow-up assessment, and dietary interview, which was performed by an expert dietitian in a subset of 37 patients to evaluate the total amount of gluten ingestion through the alimentary diary of the previous 3 days.

### 2.3. Quality of Life Assessment

Quality of life was assessed through the Psychological General Well-Being Index (PGWBI) [13], a validated questionnaire for adults and paediatric patients [14] comprehensive of 22 items, each with 0–5 score, producing a self-perceived evaluation of psychological well-being expressed by a summary score. The results of the questionnaire were grouped as follows: score 0–18 = excellent, 19–37 = fair and ≥ 38 = low QoL. As adopted for the compliance questionnaire, patients aging ≥ 12 years were asked to answer autonomously, while for patients aging < 12, guardians completed PGWBI together with their children.

In order to further assess QoL, the presence of any gastrointestinal symptom was registered using GSRS (Gastrointestinal Symptom Rating Scale), a 15 item internationally validated questionnaire.

### 2.4. Statistical Analysis

Continuous variables were screened for normal distribution and transformed to reduce skewness, when required. Student *t*-test and Chi square were adopted to compare means or proportions. ANOVA was used to compare means. A *p* value < 0.05 was considered to be statistically significant. Data were analysed with the SPSS statistical package version 25.

## 3. Results

### 3.1. Assessment of Adherence to GFD

Diet adherence resulted excellent in 95 (59.4%), fair in 46 (28.8%) and low in 19 (11.9%) of the total 160 celiac children.

In order to assess whether this difference was evident between children and adolescents, we compared levels of diet adherence between the two groups of patients (2–11 years old and ≥ 12–20 years old); better diet adherence was achieved by younger patients X^2^ = 11.9, *p* = 0.003 (Table 2).

Anti-transglutaminase antibodies turned out to be above the cut-off values in 19/160 (11.9%) and no statistically significant correlation was evident between serology and the questionnaire results.

The dietary interview of the 37 random sample revealed the ingestion of significant amount of gluten only in 2/37 (5.4%) children (Table 3). For the vast majority of celiac children surveyed (35/37, 94.6%), no significant consumption of gluten could be found, in accordance with the short questionnaire, which estimated that 141 cases (88%) reported an excellent or fair adherence to the GFD. The single case consuming about 800 mg gluten/day was classified at low adherence to the diet by the questionnaire. The other consuming a small quantity of gluten (around 200 mg) was classified as moderately adherent to the diet.

### 3.2. Factors Influencing Diet Adherence

#### 3.2.1. Sociocultural Factors

In order to assess whether diet adherence may be influenced by the socio-cultural environment, we correlated compliance with parental education and job. Data about parents were available for 136 of 160 patients. Of 136 mothers, 102 (75%) had lower school education while 34 (25%) received higher education (High school or University). Similarly, 103 (75.7%) of 136 fathers received lower school education, while 33 (24.3%) higher education. Surprisingly, we found an inverse correlation between the mothers’ education level and score of diet adherence (X^2^ = 6.86, *p* = 0.032). In particular, the percentage of children with excellent adherence was 52.9% and 62.7%, in mothers with high and low education, respectively. When considering fathers, however, this relation was inverted (X^2^ = 5.52, *p* = 0.066): excellent adherence was achieved in 66.7% and 58.3% of patients with fathers of high and low education level, respectively.

In order to investigate whether the poorer diet adherence achieved by children with highly educated mothers could be explained by the amount of time spent at home by “working mothers”, we analysed the relation between diet adherence and parents’ job. We did not observe a significative correlation between mothers’ or fathers’ job and children grade of adherence to GFD.

#### 3.2.2. Factors Related to the Patient

##### Gender and Grade of Education

According to our analysis, there was no statistically significant difference in diet adherence between the two genders.

Adherence to GFD seems to be related to the grade of education, with excellent scores dropping from 70.7% at elementary school to 60.5% at middle school till 47.5% at high school (X^2^ = 6.57; *p* = 0.037). Obviously, the education grade is strongly related to age in our setting (Spearman rho = 0.927) (Table 4).

Interestingly, we observed important differences by gender: while girls became less compliant from elementary until high school, boys showed a steadier grade of diet adherence throughout the education stages. The percentage of girls with excellent adherence, indeed, dropped significantly from 75% at elementary school to 60.7% at middle school till 40.5% at high school (X^2^ = 11.2; *p =* 0.02). Boys, instead, showed stable levels of diet adherence throughout the school period, being the percentage of patients with excellent adherence 63.6%, 60% and 59.1%, respectively at elementary, middle, and high school (Table 5).

#### 3.2.3. Factors Related to the Disease

##### Age at Diagnosis and Length of Follow Up

In our patients, the average age at diagnosis was 7.2 ± 4.6 years. According to our analysis, the age at diagnosis did not affect significantly adherence to GFD (ANOVA F = 0.228, *p =* 0.634). Average length of follow up was 4.55 ± 3.95 years (range 1–18). Interestingly, we observed an inverse correlation between grade of adherence and average length of follow up (ANOVA F test for linearity = 6.82, *p =* 0.01): average length of follow up went, indeed, from 3.88 years in patients with excellent adherence to 5.52 years in those with lower adherence.

##### Diagnostic Approach

Of all the patients, 132 were diagnosed by performing intestinal biopsy, while in 27 (16.8%) a biopsy sparing approach according to 2012 ESPGHAN guidelines [15] was adopted. Interestingly, statistically significative higher score of adherence to GFD were observed in children diagnosed with biopsy, 60% excellent if diagnosed with biopsy versus 42% if diagnosed without it (X^2^ = 6.66, *p* = 0.036).

### 3.3. Outcomes Related to Diet Adherence

#### 3.3.1. Quality of Life

Gastrointestinal symptoms were reported only by 6 (3.7%) of our patients, while 154 (96.3%) were asymptomatic when evaluated with GSRS. All the patients underwent PGWBI questionnaire, exploring Quality of Life. There was no correspondence between QoL and symptoms (GSRS), since 5 (83.3%) of the 6 patients who reported the presence of gastrointestinal symptoms also reported excellent QoL. According to PGWBI, excellent QoL was achieved by 125 of 160 (78.1%) children and was more frequently reported by girls (81.2%) than boys (72.9%) (X^2^ = 10.7, *p* = 0.005). On the other hand, an improvement in QoL along school career was evident for boys and not for girls (X^2^ = 10.3, *p* = 0.03). Our analysis showed that a higher grade of diet adherence assessed by the questionnaire significatively correlated with a higher score of QoL (X^2^ = 12.3, *p* = 0.015).

#### 3.3.2. Growth

Weight was ≤10° pct (percentile) for sex and age in 35.8% of the children and ≥90° percentile in 5%. A general trend toward lower percentiles was observed also for height, that was ≤10° pct in 26.1% and ≥90° pct only in <2% of the patients. BMI showed, instead, a nearly normal distribution being ≤10° pct in 13.8% and ≥90° in <10% of our cohort. We found a correlation between better compliance at the questionnaire and healthier growth percentiles (Figure 1): in patients with excellent diet adherence, indeed, average percentiles of weight, height and BMI were 44.89, 41.91 and 55.28, respectively; in those with fair adherence they were 35.09, 38.91 and 50.80, respectively; in those with poor adherence they were even lower, dropping to 32.72, 37.72 and 45.00, respectively. The linear trend was significative for weight (ANOVA F for linearity = 3.99 *p* = 0.047); however, it did not reach statistical significance.

## 4. Discussion

Though gluten exposure may often be unintentional, due to cross-contamination, the reasons for diet failure among paediatric patients are numerous and tend to vary with the different stages of growth. In this domain, the evaluation of diet adherence becomes a difficult task for the clinician, who should both identify the influencing factors for the different classes of age (education, socio-cultural environment) and consider outcomes (growth, morbidity, QoL).

Despite the importance and the complexity of this issue, the management of compliance with GFD is not codified. In clinical practice, serology and dietary interview are mostly used; nevertheless, both these methods show important limits. Anti-TG2 titers are, indeed, not sensitive enough to detect transgression of GFD [16]; dietary interview is not standardized and requires a close relationship between the examiner and the patient, which becomes particularly challenging in adolescence. Recently, new techniques have been developed to detect gluten consumption by assessing the excretion of gluten immunogenic peptides (GIP) [6] in stool (until 2 weeks after gluten ingestion) and urine (till 72 h). Though non-invasive and useful to identify unaware gluten ingestion in symptomatic or asymptomatic patients, the role of GIP tests in the management of CD patients is still not defined.

In this study, we aimed to evaluate the role of the questionnaire, rapid and simple to administer, in the assessment of adherence to GFD. We therefore adapted Leffler questionnaire [10] to paediatric patients with the following results: of the total 160 patients, 59.4% showed excellent adherence, while 28.8% and 11.9% showed fair and low adherence, respectively. The fact that 40.6% of our patients showed problems in adherence to GFD is consistent with data reported by other studies [2,3,17]. Our questionnaire did not correlate with anti-TG2 titers (positive in 11.9%), which also resulted negative in patients who reported problems in diet adherence with the questionnaire, confirming the scarce sensitivity of serology. Moreover, the questionnaire results did not correlate with dietary interview.

In conclusion, despite the rapidity and good sensitivity in identifying problems of adherence to GFD, the questionnaire may not be sufficient to evaluate compliance alone, in absence of other objective tools (prospective dietary enquiry, stool analysis, etc). However, for its characteristics, the questionnaire should be included in follow up protocols as a screening tool to identify patients who need an extensive evaluation by expert dietitian.

A secondary aim of this study was the identification of factors influencing adherence to GFD. The knowledge of these factors is essential to recognize children who need personalized care during follow up in order to prevent diet failure. Since it has been described that adherence to GFD may be influenced by the parents’ knowledge and awareness of the disease [5], we analysed parents’ education and job among environmental factors. Additionally, financial status may influence diet adherence, though this is mostly strongly connected with grade of education and job. Interestingly, higher scores of diet adherence were achieved by children with mothers with low levels of education and highly educated fathers. We hypothesized this tendency to be connected with a lack of time spent at home by highly educated mothers who may have more demanding jobs; however, although we registered lower adherence when mothers worked as freelancers or employees than as housekeepers, in contrast with fathers, this difference was not statistically significant and further data are needed to explain this tendency. Surely the current social dynamics, with the increase of mothers’ employment, call for a reorganization of domestic roles, with fathers increasingly involved in the management of GFD. In this scenario, also the physician may play an important role, involving more and more fathers in the clinical path of their children.

Analysing individual influencing factors, we observed how adherence to GFD tends to decrease along school career. It seems to be related to the age of patients. High school, which sees a decrement of diet adherence, corresponds with adolescence, a delicate moment characterized by lower parents’ decreased control and acquisition of autonomy in the diet management. This observation, which is in accordance with other studies [2,5,16,18], highlights the necessity of individualized treatment in adolescence, to improve autonomy and prepare the transition process.

The observation of negative correlation between diet adherence and length of follow up may also depend on age, as patients with longer follow up are mostly adolescents. However, another recent study [16] reported a significant worsening of food adherence in the same individual after 24 months of follow up. Therefore, the explanation could be a consequence of a drop of vigilance and awareness of the disease after years of wellness, the absence of symptoms after gluten consumption being the most common cause of teenage diet failure [2]. This calls for a constant renewal of information about diagnosis, GFD and risks related to diet-failure, above all in adolescence.

In our cohort, patients diagnosed with a biopsy sparing approach, according to 2012 ESPGHAN guidelines, were 27 (16.8%). Excellent compliance with GFD was, indeed, achieved by 79/132 (59.8%) vs. 15/27 (55.6%) patients who did and did not undergo biopsy respectively. These data suggest a negative correlation between compliance and biopsy sparing approaches, likely due to the lack of awareness of the disease in children who undergo a simplified diagnostic approach. However, further data on the basis of new 2019 ESPGHAN guidelines [19] are needed to confirm this hypothesis.

Another secondary aim of this study was to investigate the different outcomes related to diet adherence. Quality of life was assessed trough a validated questionnaire (PGWBI) [14]. No correspondence was found between PGWBI results and the presence of gastrointestinal symptoms (assessed throughout GRSR questionnaire), confirming the complexity of the concept of wellness, which is not only dependent on gastrointestinal symptoms. A strong correlation was observed between better compliance and higher Quality of Life at PGWBI; this suggests how dietary adherence strongly contributes to the general wellness and QoL, as reported in other studies [20,21]. This relation is confirmed by the observation that boys, who show a steadier compliance along the school career, also display improvement in QoL if compared with girls, who show a worsening of compliance with lack of improvement in QoL along the school career.

Anthropometric analysis of our cohort revealed a distribution of weight and height toward lower percentiles if compared with other studies [22] and with growth standards. Since children with different length of follow up were included in this study, for some, the catch up growth process may not have been completed yet [23,24]. Observing BMI distribution, however, it was much more similar to normal distribution, being ≤10° percentile for sex and age in 13.8% and ≥90° in <10% of the patients. Moreover, we did not observe high percentages of overweight and obese children, as reported in other studies [25,26]. Though in GFD it has been described higher intake of fat and a lower intake of fibres [26,27], predisposing to obesity [25,28], the influx of the Mediterranean diet in our cohort from southern Italy could represent a protecting factor against excessive use of commercial gluten free products and weight gain. Finally, a direct correlation was found between growth and diet-adherence (Figure 1), as reported in other studies [29,30], outlining adherence to GFD as the most important resource to guarantee CD patients normal development and wellness.

## 5. Conclusions

In conclusion, according to this study, the questionnaire shows a good sensitivity in detecting problems in diet adherence, and could be a useful screening tool in the assessment of compliance with GFD. Another point of strength of this study is that it recognises influencing factors (mothers’ education, children grade of education, length of follow up, diagnostic approach) which may identify children at risk of poor adherence to GFD, who require a personalized approach during the follow up. This calls for the implementation of specific and personalized strategies aimed to improve compliance, in the context of standardized protocols for the follow-up of coeliac patients.

The role of the expert dietitian should be implemented, as dietary interviews still represent the most accurate method of assessing diet adherence. A weakness of our study was that only 37 of 160 patients underwent a dietary interview, and this calls for an extension of the sample size to raise the accuracy of this analysis.

Another limitation was that the study population included individuals from 2–20 years of age, which is a wide range. However, the analysis of this range shows how diet adherence changes throughout the length of follow up, the school career and growth.

## Figures and Tables

**Figure 1 nutrients-12-03802-f001:**
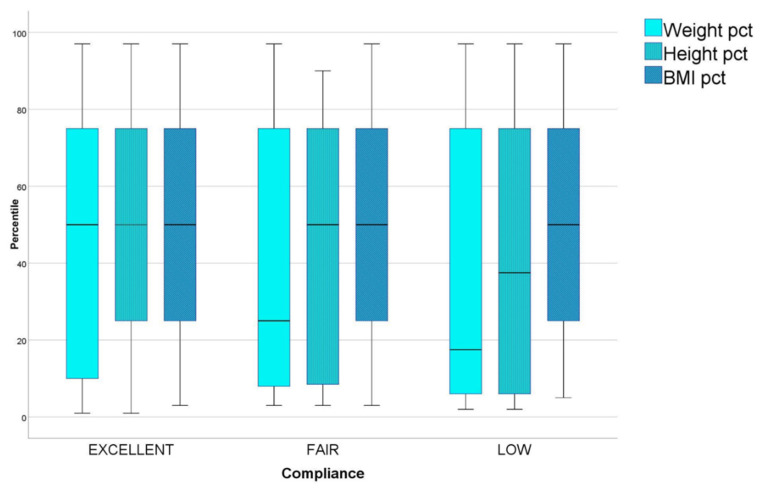
Average weight, height and BMI percentiles in patients with excellent, fair and low adherence to GFD.

**Table 1 nutrients-12-03802-t001:** Questionnaire for the assessment of adherence to gluten-free diet (GFD) in paediatric patients.

	0	1	2	3
Which is your average gluten intake?	Less than 3 times/year	Less than 6 times/year	Monthly	1–2 times/week
When dining outside	You go to coeliac only friendly restaurants or extensively enquire about food preparation	You make useful questions on food preparation, but you don’t look for coeliac-friendly restaurants	You make few useful questions on food preparation, but you don’t look for coeliac-friendly restaurants	You do not enquire about food preparation
Do you consider accidental gluten exposures important to your health?	Yes	Unsure	A little important	Not at all important
Do you avoid all potential cross-contact in the kitchen?	All of them	Most of them	Some of them	None of them
Do you avoid gluten during all social, familiar and scholar activities?	All of the time	Most of the time	Some of the time	Never
Do you consider yourself ill?	No	Seldom	Some of the time	Yes
Do you check all medications, nasal products and vitamins are gluten free?	All of the time	Most of the time	Some of the time	Never
Other	You carefully read food label		You do not carefully read food labels/Mental retardation or behaviour problems	You have been bothered by low energy levels during the past 4 weeks

**Table 2 nutrients-12-03802-t002:** Grade of diet adherence in 2 different groups of age (2–11 and 12–20 years old).

			Age		Total
			2–11 Years	12–20 Years	
Adherence	Low	Number of patients	2	16	18
		Percentage	3%	17.6%	11.9%
	Fair	Number of patients	16	30	46
		Percentage	23.9%	33%	28.8%
	Excellent	Number of patients	49	45	94
		Percentage	73.1%	49.5%	59.4%

Chi Square (X^2^) = 11.9, *p* = 0.003.

**Table 3 nutrients-12-03802-t003:** Adherence to GFD assessed through the questionnaire compared with serology (Anti-TG2) and dietary interview.

**Adherence**		**Questionnaire**		**Anti-TG2**		**Dietary Interview**	
	Number of patients	Low	19	Positive	19	Fair/Low	2
	Percentage		11.9%		11.9%		5.4%
	Number of patients	Fair	46				
	Percentage		28.8%				
	Number of patients	Excellent	95	Negative	141	Excellent	35
	Percentage		59.4%		89.1%		94.6%
	Number of patients	Total	160	Total	160	Total	37
	Percentage		100.0%		100.0%		100.0%

**Table 4 nutrients-12-03802-t004:** Adherence to GFD tends to decrease with increasing age.

			Age at Enquiry					Total
			2–4	5–9	10–14	15–20	>20	
Adherence	Fair/Low	Number of patients	3	12	25	22	2	64
		Percentage	20.0%	34.3%	41.7%	50.0%	50.0%	40.5%
	Excellent	Number of patients	12	23	35	22	2	94
		Percentage	80.0%	65.7%	58.3%	50.0%	50.0%	59.5%

Chi Square (X^2^) for trend = 4.714; *p* = 0.030.

**Table 5 nutrients-12-03802-t005:** Difference in diet adherence between boys and girls throughout the education stages.

Gender				School			Total
				Elementary	Middle	High	
Girls	Adherence	Fair/Low	Number of patients	9	11	22	42
			Percentage	25.0%	39.3%	59.5%	41.6%
		Excellent	Number of patients	27	17	15	59
			Percentage	75.0%	60.7%	40.5%	58.4%
Boys	Adherence	Fair/Low	Number of patients	8	6	9	23
			Percentage	36.4%	40.0%	40.9%	39.0%
		Excellent	Number of patients	14	9	13	36
			Percentage	63.6%	60.0%	59.1%	61.0%

Girls: Chi Square (X^2^) = 11.2 *p* = 0.02; Boys: Chi Square (X^2^) = 0.3 *p* = 0.9.

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
