# Peer review of "Adherence to Gluten-Free Diet in Coeliac Paediatric Patients Assessed through a Questionnaire Positively Influences Growth and Quality of Life"

_nutrients, 2020, doi:10.3390/nu12123802_

Round 1

Reviewer 1 Report

Thank you for asking me to review this paper.  It will need some editing for grammatical errors and english usage.  

It describes the use of a refined 'Leffler' questionnaire to identify potential issues with adherence to a gluten free diet in a largely paediatric population.  Interesting features that are highlighted are the lack of correlation of findings from the questionnaire with TTG titres (as many have demonstrated) and with dietary review.

There are a few features that could be enhanced.  Most importantly the study population comprises individuals from 2-20 years of age.  This is an extremely wide range of different ages and competencies and comparing the attitudes of a 20 year old to a 10 year old for instance may dilute the significance of key findings - particularly with regard to growth characteristics.  Given that the mean age was around 7 years, it might be best to either limit the upper age or separate the analysis (perhaps <12 and >12 years of age given that this is the age at which the authors chose to allow participants to complete their own questionnaires.  I suspect that the paper and the data analysis would be stronger by this approach.

Secondly, the dietary interview was only carried out in 37 individuals.  A small number, but nevertheless the questionnaire should be directly compared within this group and the findings shown.  It is unclear on what grounds the lack of correlation between dietary review and questionnaire is assessed - is this purely by the fact that a smaller proportion of the dietary review group were thought to be non adherent than the questionnaire group?  Perhaps those found to be poorly adherent on dietary review were also within the most concerning group for the questionnaire? how many were in the 'excellent' group or the 'fair' groups based on the questionnaire cut-offs.  These data need to be shown, as do the data showing lack of correlation with TTG titres.  I would recommend adding a table and perhaps dividing into questionnaire based groups - excellent, fair and low adherence to demonstrate the proportions and the p values throughout the data presentation.

One linguistic subtlety of the authors should be aware is that coeliac professionals currently try to refer to dietary 'adherence' rather than 'compliance' which has overtones of compulsion rather than choice.   

Author Response

Reviewer 1:

Point 1: Most importantly the study population comprises individuals from 2-20 years of age.  This is an extremely wide range of different ages and competencies and comparing the attitudes of a 20 years old to a 10 years old for instance may dilute the significance of key findings - particularly with regard to growth characteristics.  Given that the mean age was around 7 years, it might be best to either limit the upper age or separate the analysis (perhaps <12 and >12 years of age given that this is the age at which the authors chose to allow participants to complete their own questionnaires.  I suspect that the paper and the data analysis would be stronger by this approach.

Response 1: Thanks for the suggestion. We analyzed diet adherence data by dividing the population into two groups (<12 years and> 12 years of age). The data confirm that the best compliance is in younger children (see text page 4, lines 104-106, and table 2.

Point 2: Secondly, the dietary interview was only carried out in 37 individuals.  A small number, but nevertheless the questionnaire should be directly compared within this group and the findings shown.  It is unclear on what grounds the lack of correlation between dietary review and questionnaire is assessed - is this purely by the fact that a smaller proportion of the dietary review group were thought to be non adherent than the questionnaire group?  Perhaps those found to be poorly adherent on dietary review were also within the most concerning group for the questionnaire? how many were in the 'excellent' group or the 'fair' groups based on the questionnaire cut-offs.  These data need to be shown, as do the data showing lack of correlation with TTG titres.  I would recommend adding a table and perhaps dividing into questionnaire based groups - excellent, fair and low adherence to demonstrate the proportions and the p values throughout the data presentation.

Response 2: We welcomed the reviewer's suggestion and analyzed the data in the group of 37 children with nutritional interview. The very small numbers unfortunately do not allow us to make definitive conclusions, but the two children with traces of gluten in the diet both fall into the category with fair/low adherence. We compared also adherence with anti-tTG2 (see text page 4 lines 109-115, and table 3)

Point 3: One linguistic subtlety of the authors should be aware is that coeliac professionals currently try to refer to dietary 'adherence' rather than 'compliance' which has overtones of compulsion rather than choice.   

Response 3: The text has been modified according to the suggestion

Reviewer 2 Report

Pedoto D et al. studied adherence to a gluten-free diet and associated factors in 160 pediatric celiac disease patients. Adherence was evaluated with modified questionnaire and in 37 patients also with a dietary interview. The questionnaire results did not correlate with celiac autoantibodies or dietary interview but gave a broader view about the adherence and its difficulties. Results are clinically relevant, and manuscript is mainly well-written and easy to read. However, results should be reported more precisely (e.g. in table format) and possibility of cofounding factors should be taken account. Some of the methods also need more specific description. Limitations of the study should be listed in Discussion.

Title

  • Consider including the main results in the title.

Abstract

  • Lines 19-20: Sentence “Compliance inversely correlated with average length of follow-up” is a little bit difficult to understand.

Introduction

  • Please provide references to the following sentences
    • Lines 33-34: “…representing the most important factor influencing prognosis in terms of morbidity and quality of life (QoL).
    • Line 37: “…with significant consequences on growth and QoL.”
  • Line 42: Reference #8 should be marked similarly to other references.
  • Consider moving the description of GIP in discussion section as the method is not used in the present study and is introduced mainly to provide a wider aspect to the topic.
  • Line 61: Presence of other autoimmune diseases could also affect the dietary adherence so these could be also affecting factors not just outcomes.
  • Lines 63-65: Consider using term “associated with” instead of repeating “influenced by” twice explaining the possibly two-way effects.

Material & Methods

  • How was the dietary interview performed?
  • What method was used to measure TG2 antibodies and when were the antibodies measured regarding to dietary interview and questionnaires?
  • Lines 89-94: Has PGWBI questionnaire previously been used in pediatric patients?
  • Statistical analyses should be more precisely described.
  • Ethical aspects of the study should be mentioned.

Results

  • Describing the comparison of different groups (e.g. those with excellent vs fair/low dietary adherence) also in tables and/or figures could help to outline the results.
  • Is it possible to further compare mothers with high school vs university education? Were some of the mothers still students which could affect their financial situation and therefore gluten-free diet negatively?
  • Did the results differ between the patients who had responded themselves compared to those whose parents had responded?
  • Possibilities of cofounding factors explaining the findings should be considered. Was there e.g. difference in current age or other characteristics between those diagnosed with biopsy sparing vs conservative methods? Or between parents with different education?

Discussion

  • Consider discussing the possible significance of family’s financial situation to the dietary compliance.
  • Lines 264-266: Would it be possible to provide some suggestions about strategies to improve compliance based on these results?
  • Consider including a chapter with strengths and limitations of the study.

Author Response

Reviewer 2:

Pedoto D et al. studied adherence to a gluten-free diet and associated factors in 160 pediatric celiac disease patients. Adherence was evaluated with modified questionnaire and in 37 patients also with a dietary interview. The questionnaire results did not correlate with celiac autoantibodies or dietary interview but gave a broader view about the adherence and its difficulties. Results are clinically relevant, and manuscript is mainly well-written and easy to read. However, results should be reported more precisely (e.g. in table format) and possibility of cofounding factors should be taken account. Some of the methods also need more specific description. Limitations of the study should be listed in Discussion.

Title

  • Consider including the main results in the title.
  • The title has been modified according to the suggestion

 Abstract

  • Lines 19-20: Sentence “Compliance inversely correlated with average length of follow-up” is a little bit difficult to understand.
  • The sentence has been modified according to the suggestion (line 20-21)

Introduction

  • Please provide references to the following sentences:

Lines 33-34: “…representing the most important factor influencing prognosis in terms of morbidity and quality of life (QoL). We added reference

Line 37: “…with significant consequences on growth and QoL.” We added reference

  • Line 42: Reference #8 should be marked similarly to other references. Done
  • Consider moving the description of GIP in discussion section as the method is not used in the present study and is introduced mainly to provide a wider aspect to the topic. We moved the GIP’s description in the discussion as suggested (page 7, lines 217-221)
  • Line 61: Presence of other autoimmune diseases could also affect the dietary adherence so these could be also affecting factors not just outcomes. We welcome the remark and modified the text (see page 2, line 52).
  • Lines 63-65: Consider using term “associated with” instead of repeating “influenced by” twice explaining the possibly two-way effects. Done

 Material & Methods

  • How was the dietary interview performed? 37 children underwent the dietary interview (page 2, line 81)
  • What method was used to measure TG2 antibodies and when were the antibodies measured regarding to dietary interview and questionnaires? Anti-TG2 test were measured by ELISA (Eurospital, Trieste, Italy) or by chemiluminescence (Delta) at time of follow-up assessment (page 2, line 80-81).
  • Lines 89-94: Has PGWBI questionnaire previously been used in pediatric patients? Yes, we add a references (14)
  • Statistical analyses should be more precisely described. We tried to describe more precisely statistical methods (page 4, line 97-98).
  • Ethical aspects of the study should be mentioned. We mentioned it in page 2, lines 67-68. 

Results

  • Describing the comparison of different groups (e.g. those with excellent vs fair/low dietary adherence) also in tables and/or figures could help to outline the results. Thanks for the suggestion, we added four tables, that show results based on the level of adherence to the diet found (see table1, 2, 3,4).
  • Is it possible to further compare mothers with high school vs university education? Were some of the mothers still students which could affect their financial situation and therefore gluten-free diet negatively? No mother reported to be still student at the moment of the visit
  • Did the results differ between the patients who had responded themselves compared to those whose parents had responded? We divided the results on the basis of age, because children over the age of 12 responded independently to the questionnaires, while for those under the age of 12 it was the parents who answered (see tab.2, page 4)
  • Possibilities of cofounding factors explaining the findings should be considered. Was there e.g. difference in current age or other characteristics between those diagnosed with biopsy sparing vs conservative methods? Or between parents with different education? We analysed some possible cofounding factor, as suggested: parental education effects is reported in page 5, lines 128-141; diagnostic approach for the diagnosis is mentioned in page 6, lines 174-178.

Discussion

  • Consider discussing the possible significance of family’s financial situation to the dietary compliance. Thanks for the suggestion, we add a sentence in the discussion (page 8, lines 239-240).
  • Lines 264-266: Would it be possible to provide some suggestions about strategies to improve compliance based on these results? Our proposal to improve compliance with the diet is to include a dietician in the follow-up of celiac children. The role of the expert dietitian should be implemented, as dietary interview still represents the most accurate method to assess diet adherence (page 9, lines 299-300)
  • Consider including a chapter with strengths and limitations of the study. We welcome the suggestion and added a chapter on strengths and limitations (page 9, lines 294-305).